# Association between vitamin D receptor gene polymorphism and essential hypertension: An updated systematic review, meta-analysis, and meta-regression

**Iwan Dakota[1], Muhamad Fajri Adda'i[1], Rido Maulana[1], Ignatius Ivan[2], Renan Sukmawan[1], Bambang Widyantoro[ID][1] ***

1 Department of Cardiology and Vascular Medicine, Faculty of Medicine, National Cardiovascular Centre Harapan Kita, Universitas Indonesia, Jakarta, Indonesia, 2 School of Medicine and Health Sciences, Atma Jaya Catholic University of Indonesia, Jakarta, Indonesia

* bambang.widyantoro@pjnhk.go.id

**Data Availability Statement:** All relevant data are within the manuscript and its Supporting Information files.

## Abstract

The association between Vitamin D Receptor (VDR) gene polymorphisms and essential hypertension (EH) remains controversial. We searched databases (Cochrane Library, EBSCO, EMBASE, LILACS, ProQuest, PubMed, Science Direct, Springer) for studies on VDR gene polymorphisms and EH until May 30, 2024, following PRISMA guidelines. Rev-Man 5.4.1 provided pooled odds ratio (OR) under Hardy-Weinberg Equilibrium based on allele, additive, dominant, and recessive genetic models. Meta-regression was performed using Comprehensive Meta Analysis V3. Twenty-two studies from thirteen countries were analyzed. The recessive model suggested lower EH risk in individuals with the recessive allele (bb) of BsmI (OR: 0.81; 95%CI, 0.69 to 0.94, p = 0.007; $I^2$ = 35%, p = 0.13). No significant associations were found for FokI, ApaI, and TaqI polymorphisms. Methodological quality significantly influenced EH risk associated with the FokI polymorphism across allele, additive, and dominant models (All p<0.0005). Male proportion influenced EH risk in the additive model for the FokI polymorphism (p = 0.0235), while age impacted risk in the recessive model (p = 0.0327). FokI polymorphism's influence on EH risk varies by sex, age, and study quality. BsmI polymorphism is independently associated with lower EH risk in recessive homozygotes, with no significant associations found for ApaI and TaqI polymorphisms.

## Introduction

Essential hypertension (EH) is still a global health problem. In 2025, it was estimated that 1.56 billion people globally would have EH [1]. Those populations with EH have a higher risk of coronary artery disease, cerebrovascular disease, and heart failure [2]. Recently, genetic polymorphism of Vitamin D receptor (VDR) has been linked to EH development in 30–50% cases [3]. Beyond essential hypertension (EH), VDR gene polymorphisms have also been linked to

**Funding:** The author(s) received no specific funding for this work.

**Competing interests:** The authors have declared that no competing interests exist.

**Abbreviations:** 1,25 [OH]2 D, 1,25-dihydroxyvitamin D; 3'UTR, 3' untranslated region; CIs, confidence intervals; DBP, diastolic blood pressure; EBSCO, Elton B. Stephens Company; EMBASE, Excerpta Medica Database; EH, essential hypertension; HWE, Hardy-Weinberg equilibrium; IQR, interquartile ranges; LILACS, Literatura Latino Americana e do Caribe em Ciências da Saúde; mRNA, messenger ribonucleic acid; NOS, Newcastle-Ottawa Scale; OR, odds ratio; PCR, polymerase chain reaction; PRISMA, Preferred Reporting Items for Systematic reviews and Meta-Analyses; PROSPERO, International Prospective Register of Systematic Reviews; REM, random effect model; SBP, systolic blood pressure; SD, standard deviation; SE, standard of error; SNP, single nucleotide polymorphism; VDR, vitamin D receptor.

other common diseases such as osteoporosis [4], type 2 diabetes mellitus [5], and various auto-immune conditions, including multiple sclerosis [6] and systemic lupus erythematosus [7]. BsmI, FokI, ApaI, and TaqI polymorphisms have demonstrated varying degrees of association with these diseases due to their influence on vitamin D metabolism, immune modulation, and cellular differentiation [8]. However, the specific role of VDR polymorphisms in EH has not been fully elucidated, which justifies our focused analysis on EH in this study. Additionally, in cardiovascular metabolism, 1,25-dihydroxyvitamin D (1,25 [OH]2 D) binds to the VDR, a transcription factor in the aortic endothelium, and smooth muscle vascular cells to exercise its biological effects on myocardiocytes [9–11].

The most investigated polymorphisms of VDR are FokI (rs2228570, C>T) and BsmI (rs1544410, G>A). FokI polymorphisms are made up of a T to C transition that changes the start codon and the length of the vitamin VDR protein. Compared to the polymorphic form that contains the variant C (allele F), the variant T (allele f) in FokI, produce the longer total protein with 427 amino acids which have a lower level of biological activity. VDR with the FokI CC genotype (FF) that has shorter a total protein of 424 amino acids resulting in higher VDR protein activity when compared to VDR with the CT (Ff) or TT (ff) genotypes [12–16]. This is because CC are more likely to be found in the VDR gene. The truncated protein, namely the 424 amino acid protein, found in individuals with the FF genotype is believed to contribute to the onset of EH by enhancing the production of renin and angiotensin II. There-fore, dominant homozygous (FF) and heterozygous (Ff) have a 2.2-fold increased chance of developing EH compared to recessive homozygous (ff) [17].

The BsmI polymorphisms is an A to G nucleotide substitution that can be found in intron 8 and has an effect on the stability of the transcript [18]. These polymorphisms have been linked with an increased likelihood of developing EH [18]. Therefore, BsmI GG genotype (BB) might also contribute to EH. Conversely, the ApaI and BsmI polymorphisms, traditionally recog-nized for their involvement in modulating messenger ribonucleic acid (mRNA) stability and protein translation of VDR, were postulated to exert an influence on vitamin D functionality, consequently affecting plasma renin activity and blood pressure levels [19, 20].

The polymorphisms chosen for this study (BsmI, FokI, ApaI, and TaqI) are the most commonly studied and clinically relevant variants in the VDR gene [21]. Previous study has identified these polymorphisms as the genetic variants associated with blood pressure regu-lation through mechanisms involving the renin-angiotensin system (RAS) and vascular function [21]. Other polymorphism, such as Tru9I (rs757343), has been studied in relation to vitamin D metabolism, but evidence for their direct involvement in EH is still limited [22]. Additionally, polymorphisms such as Cdx2 (rs11568820) are known to affect the VDR gene promoter region, but they are more commonly associated with conditions like bone density or inflammatory diseases rather than EH [23]. Given the existing evidence and the higher prevalence of studies on these polymorphisms, we focused our analysis on the BsmI, FokI, ApaI, and TaqI variants.

Previous meta-analysis of 7 studies had investigate association between VDR gene polymor-phism and susceptibility of EH [21]. Results showed that AA genotype (bb) frequency was decreased in EH patients compared with healthy controls indicating that AA genotype (bb) reduce susceptibility to EH [21]. Meanwhile, this meta-analysis did not find significant correla-tion between FokI polymorphism and susceptibility to EH [21]. Emerging evidences revealed other single nucleotide polymorphism (SNP) that may play role in increasing the susceptibility of EH along with additional studies that may modify previous results [24]. In accordance with the consensus for updating systematic review [25], our aim is to elucidate the relationship between VDR gene polymorphisms and the risk of EH.

## Methods

### Protocol, registration, and ethics statement

The protocol of this systematic review was registered at the International Prospective Register of Systematic Reviews (PROSPERO) with registration number CRD42023409215. This systematic review followed the Preferred Reporting Items for Systematic reviews and Meta-Analyses (PRISMA) 2020 guideline for reporting [26]. This study is a systematic review of previously published studies and does not involve direct research on human or animal subjects. As such, ethical approval and informed consent were not required.

### Data source

We carried out a comprehensive search to identify all published observational studies, that investigate the relationship between VDR gene polymorphisms (FokI, BsmI, ApaI, TaqI and other related SNP) and risk of EH. Cochrane library, Elton B. Stephens Company (EBSCO), Excerpta Medica Database (EMBASE), Literatura Latino Americana e do Caribe em Ciências da Saúde (LILACS), ProQuest, PubMed, Science Direct, and Springer databases were searched from inception until May 30, 2024. Grey Literature Report and OpenGrey database were additionaly searched for grey literature. Manual searching was also done in Google Scholar.

### Search strategy, study eligibility, and study selection

Our search strategy used medical subject headings terms with title or abstract searching by complementing keywords combination of the following: hypertension, vitamin D, polymorphism, gene variation.

The inclusion criteria for this meta-analysis were rigorously defined to ensure the selection of high-quality, relevant studies. First, we included studies that investigated adult populations, specifically individuals aged 18 years or older, who had been clinically diagnosed with EH or were undergoing treatment with antihypertensive medications. Diagnosis of EH was based on standard clinical guidelines, such as sustained systolic blood pressure (SBP) levels of 140 mmHg or higher and/or diastolic blood pressure (DBP) levels of 90 mmHg or higher. To be eligible, studies needed to investigate the association between VDR gene polymorphisms, including FokI, BsmI, ApaI, TaqI, or other SNPs, utilizing validated genotyping methods such as polymerase chain reaction (PCR), PCR-restriction fragment length polymorphism (PCR-RFLP), or PCR-TaqMan assay.

Comparisons were required to include a control group consisting of normotensive participants, defined as having SBP < 140 mmHg and DBP < 90 mmHg, with no history of hypertension or antihypertensive treatment. The primary outcome measure was the risk of EH, expressed as odds ratios (OR) with 95% confidence intervals (CIs). We included only observational studies—case-control, cohort, or cross-sectional—that reported genotype frequencies adhering to Hardy-Weinberg equilibrium (HWE), with a minimum of 10 participants per genotype to ensure sufficient statistical power.

Studies were excluded if they focused on non-hypertensive populations, including children under 18 years of age and pregnant women, due to the differences in blood pressure regulation mechanisms compared to the general adult population. We also excluded studies that failed to provide adequate data on allele and genotype frequencies or that lacked effect estimates (such as OR) for EH risk. Additionally, reviews, meta-analyses, commentaries, case reports, case series, conference abstracts, and animal studies were excluded to focus solely on observational studies with primary data.

Moreover, to maintain the integrity of the genetic analysis, studies that employed non-standard genotyping methods or failed to report quality measures for genotyping—such as PCR validation or adherence to HWE—were excluded. This ensured that only studies meeting the highest standards for genotyping quality were included in the final analysis.

## Data collection

Three (3) authors (MFA, RM, and II) independently screened the literature search and retrieved the titles, abstracts, and full texts of articles that matched with our search terms. The remaining investigators (ID, BW, and RS) read full selected articles that met the requirements and provided final suggestion. Final inclusion of studies was merely based on the agreements of all investigators; the disagreement was resolved by consensus.

After retrieving final included studies, two authors independently extracted data using standardized table for the following: authors, publication year, country of origin, sample size, population characteristic (age, sex, comorbidity), baseline SBP/DBP, genotyping methods, SNP loci with p value for HWE, and outcome results along with 95% confidence intervals (CIs), standard deviation (SD), standard of error (SE), or interquartile ranges (IQRs). Authors of studies were contacted via email to request access for missing data.

## Quality assessment

Methodological quality was assessed using the Newcastle–Ottawa scale (NOS) which had the criterion of the study quality and study reporting [27]. The NOS evaluated participant selection, comparability, and outcome reporting using eight subscale items [27]. For cross-sectional studies, an adapted version of NOS was used, similar to previous study [28]. Sum of subscale item scores with a maximum of ten, was used to provide an overall assessment for cross-sectionals. A maximum score of nine was used for case-control and cohorts. Quality assessment performed by two authors and disagreement resolved by consensus.

## Meta-analysis

Meta-analysis was performed using Review Manager 5.4.1 software (The Cochrane Collaboration, Oxford, UK) [29]. We expressed effect estimates by odd ratios (OR) with 95%CI for dichotomous outcomes. A p value less than 0.05 denoted statistical significance. Statistical heterogeneity was detected by Cochrane's Q test and Higgins $I^2$ statistic [30, 31]. We performed inverse variance random effect model (REM) using DerSimonian-Laird method if there was a substantial heterogeneity ($I^2 > 50\%$ or p value < 0.1). Meta-analysis was performed for all SNPs under 4 models including allele model (Major Allele vs Minor Allele), additive model (recessive homozygous vs dominant homozygous), dominant model (recessive homozygous + heterozygous vs dominant homozygous), and recessive model (recessive homozygous vs dominant homozygous + heterozygous).

## Subgroup and sensitivity analysis

We performed subgroup analysis based on continent of origin (Asia, Europe, Africa, and America). We performed sensitivity analysis by conducting a leave-one-out meta- analysis by excluding one study to re-estimate pooled effect [32]. For additional analysis, we conducted meta-analysis including only case-control studies.

## Meta-regression

We conducted meta-regression using method of moments in order to investigate the true causes of heterogeneity that explained $I^2$ statistic high value [33]. This was performed using Comprehensive Meta Analysis V3 [34] only for meta-analysis including at least 10 studies of available covariate as suggested by Cochrane Handbook to minimize the risk of overfitting in regression models [31]. Source of potential variability were based on covariates of sex proportion (percentage of male), mean age of study sample, and methodological quality of the study.

## Publication bias

We evaluate publication bias using Comprehensive Meta Analysis V3 [34] by generating Begg's funnel plot when included studies at least 10 [31, 35]. We used Egger's test with Begg and Mazumdar's rank correlation test to confirm the results [36, 37]. We performed Duval and Tweedie's trim and fill method to correct publication bias [38].

## Results

Initial search strategy identified 2370 records from database searching with 2 additional records from manual search (S1 Table). Results were imported to Endnote X9 and with duplicates removed, there were 2174 records left for review. Title and abstracts were reviewed based on inclusion and exclusion criteria and resulting 30 reports. There were 30 full text paper to be reviewed. There were 17 reports excluded because of 4 conference abstracts, 1 commentary paper, 4 reviews, 6 studies with no genotype frequency on hypertensive group, and 2 studies of pregnant patients. There were 13 reports that included in this systematic review with additional 7 reports from previous systematic review [21] and 2 studies from manual searching. Finally, a total of 22 reports representing 22 studies will be used in this systematic review (Fig 1).

This systematic review identified 13 case-control studies [39–57], 2 cross sectional studies [58, 59], and 1 cohort study [60] from a total of 13 countries (Table 1). There were 17 studies reporting about FokI polymorphism [39, 41, 42, 44, 45, 47–55, 57, 58, 60], 9 studies reporting about BsmI polymorphism [41, 42, 47–49, 51, 53, 54, 60], 7 studies reporting ApaI polymorphism [41, 42, 47, 48, 51, 54, 57], 8 studies reporting TaqI polymorphism [40–42, 46–49, 54], and 2 studies reporting other SNPs (CYP24A1 rs2762939,CYP2R1 rs1993116 and rs10741657; CYP24A1 rs4809957 and rs6068816) [43, 56]. Additionally, 1 study report only the recessive model for BsmI polymorphism and TaqI polymorphism.[59] Genotyping methods for detecting polymorphism were varied from 11 studies using PCR with Taqman [40, 43, 45, 47, 48, 50, 52, 56–59], 10 studies using PCR with restriction fragment length polymorphism (RFLP) [39, 41, 42, 44, 49, 51, 53–55, 60], or 1 study using PCR with amplification refractory mutation system (ARMS) [46]. Population age from all studies were varied between 25 to more than 100 years old. Glocke, et al. [50] and Gussago et al. [54] include patients with more than 90 and 100 years old in case group, respectively. Other comorbidities beside EH were diabetes from 3 studies [40, 48, 52], dyslipidemia from 2 studies [45, 48], stroke from 1 study [42], obesity and metabolic syndrome from 2 studies [48, 59]. There were 15 studies with case group include hypertensive patients defined by SBP $\geq$ 140 and/or DBP $\geq$ 90 mmHg [39, 41, 43, 44, 47–49, 51, 52, 55–60]. Meanwhile, there were 7 studies with case group include hypertensive patients from all stages [40, 42, 45, 46, 50, 53, 54]. The distribution frequency of all genotype in the control group from all studies were consistent with HWE (p > 0.05).

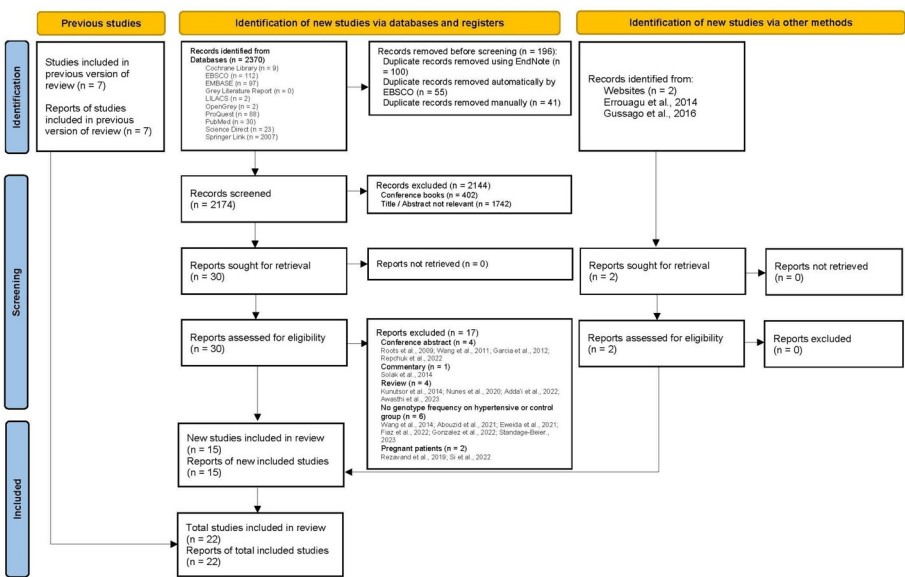

**Fig 1. PRISMA 2020 flow diagram for updated systematic reviews which included searches of databases, registers and other sources.**

## Risk of essential hypertension in patients with FokI polymorphism

We evaluated 17 studies that reported on the role of the FokI polymorphism in EH susceptibility (S1–S4 Figs) [39, 41, 42, 44, 45, 47–55, 57, 58, 60]. The pooled analysis for EH risk associated with the FokI polymorphism indicated non-significant results for the allele model (OR: 0.96; 95%CI, 0.84 to 1.10; p = 0.55), additive model (OR: 1.08; 95%CI, 0.82 to 1.44; p = 0.58), dominant model (OR: 1.00; 95%CI, 0.84 to 1.18; p = 0.99) and recessive model (OR: 1.11; 95% CI, 0.86 to 1.44; p = 0.42). Due to significant statistical heterogeneity (p < 0.1), the REM was utilized. Subgroup analyses by continent of origin for all models produced similar results (S5–S8 Figs). Notably, the allele, additive, and recessive models for the African subgroup showed significant results, but this finding was based on only one study [51]. Sensitivity analysis using a leave-one-out approach for the allele, additive, and recessive models demonstrated no changes in the pooled results (S9 Fig).

## Risk of essential hypertension in patients with BsmI polymorphism

We conducted an evaluation of nine studies that reported on the susceptibility to EH in patients with the BsmI polymorphism (Figs 2 and S10–S12) [41, 42, 47–49, 51, 53, 54, 60]. Additionally, one study focused solely on a recessive model for the BsmI polymorphism [59]. The recessive model indicated a reduced risk of EH in patients who were recessive homozygous (bb) for BsmI (OR: 0.81; 95%CI, 0.69 to 0.94, p = 0.007; $I^2$ = 35%, p = 0.13) (Fig 2). However, no significant results were found for the allele, additive, and dominant models (S10–S12 Figs). A fixed effect model (FEM) was employed for all analyses as there was no significant statistical heterogeneity. Subgroup analyses of all models based on the continent of origin also yielded no significant findings (S13–S16 Figs). Sensitivity analysis using a leave-one-out approach for the allele, additive, and dominant models showed no change in the pooled results (S17 Fig). Conversely, the leave-one-out analysis for the recessive model revealed a shift to non-significance upon the exclusion of the study by Wang et al. [60].

**Table 1. Characteristic of included studies.**

| Study, year, country | Baseline age (mean ±SD / SE / min - max) | Comorbidity | Baseline BP | Genotyping methods | SNP loci ($P_{HWE}$ for control group) | Sample Size | Results |
|---|---|---|---|---|---|---|---|
| | | | | | CASE CONTROL STUDY | | |
| Swapna et al., 2011, India [39] | EH: 55.6 (SE = 0.59) Control: 47.6 (SE = 0.66) | EH group: SBP ≥ 140 and/or DBP ≥ 90 mmHg Control: Healthy | EH group: SBP: 161.3 (SE = 1.26) DBP: 98.7 (SE = 0.81) Control: SBP: 119.0 (SE = 0.24) DBP: 80.4 (SE = 0.09) | PCR–RFLP | rs2228570 FokI $P_{HWE}$ = 0.67 | Case = 280 Male: 149 (53.2%) Female: 131 (46.8%) Control = 200 Male: 138 (69%) Female: 62 (31%) | **Fok-1** EH group: FF: 150 (53.6%) Ff: 100 (35.7%) ff: 30 (10.7%) Allele F: 400 (71.4%) Allele f 160 (28.6%) Control: FF: 68 (34.0%) Ff: 102 (51.0%) ff: 30 (15.0%) Allele F: 238 (59.5%) Allele f: 162 (40.5%) |
| Vural et al., 2012, Turkey [40] | N.R | EH group: EH with T2DM Control: Healthy | N.R | PCR–TaqMan | rs731236 TaqI $P_{HWE}$>0.05 | Case = 100 Male: 60 (60%) Female: 40 (40%) Control = 100 Male: 70 (70%) Female: 30 (30%) | **Taq-1** EH group: TT: 51 (51%) Tt: 46 (46%) tt: 3 (3%) Allele T: 148 (74%) Allele t: 52 (26%) Control: TT: 35 (35%) Tt: 49 (49%) tt: 16 (16%) Allele T: 119 (59.5%) Allele t: 81 (40.5%) |
| Glocke et al., 2013, Germany [50] | EH: 92.0 Control: 49.0 | EH group: all EH > 90 years old Control: <90 years old | All group FF: 161.50 ± 4.44 Ff: 154.31 ± 4.09 ff: 152.69 ± 6.26 | PCR–TaqMan | rs2228570 FokI $P_{HWE}$ = 0.70 | Case = 101 Control = 208 | **Fok-1** EH group: FF: 36 (35.6%) Ff: 52 (51.5%) ff: 13 (12.9%) Allele F: 124 (61.4%) Allele f: 78 (38.6%) Control: FF: 75 (36.1%) Ff: 102 (49.0%) ff: 31 (14.9%) Allele F: 252 (60.6%) Allele f: 164 (39.4%) |
| Errouagui et al., 2014, Morocco [51] | EH: 49.63 ±14.96 Control: 56.94 ±11.47 | EH group: SBP ≥ 140 and/or DBP ≥ 90 mmHg, or treated by antihypertensive drugs Control: Healthy | EH group: SBP: 147.83 ±17.78 DBP: 84.47 ±10.10 Control: SBP: 127.92 ±6.65 DBP: 73.21 ±4.14 | PCR–RFLP | rs2228570 FokI $P_{HWE}$ = 0.57 rs1544410 BsmI $P_{HWE}$ = 0.42 rs7975232 ApaI $P_{HWE}$ = 0.11 | Case = 176 Control = 177 | **Fok-1** EH group: FF: 120 (54.5%) Ff: 91 (41.4%) ff: 9 (4.1%) Allele F: 331 (75.4%) Allele f: 109 (24.6%) Control: FF: 79 (45.4%) Ff: 74 (42.5%) ff: 21 (12.1%) Allele F: 232 (66.7%) Allele f: 116 (33.3%) **Bsm-1** EH group: BB: 19 (11.4%) Bb: 71 (42.8%) bb: 76 (45.8%) Allele B: 109 (32.8%) Allele b: 223 (67.2%) Control: BB: 18 (13.2%) Bb: 57 (41.9%) bb: 61 (44.9%) Allele B: 93 (34.2%) Allele b: 179 (65.8%) **Apa-1** EH group: AA: 44 (22.1%) Aa: 113 (56.8%) aa: 42 (21.1%) Allele A: 201 (50.5%) Allele a: 197 (49.5%) Control: AA: 36 (22.8%) Aa: 89 (56.3%) aa: 33 (20.9%) Allele A: 161 (50.9%) Allele a: 155 (49.1%) |

*(Continued)*

**Table 1.** (Continued)

| Study, year, country | Baseline age (mean ± SD / SE / min - max) | Comorbidity | Baseline BP | Genotyping methods | SNP loci ($P_{HWE}$ for control group) | Sample Size | Results |
|---|---|---|---|---|---|---|---|
| Jia et al., 2014, China [52] | EH: 60.7±11.2 Control: 58.1 ±11.0 | EH group: SBP ≥ 140 and/or DBP ≥ 90 mmHg, or treated by antihypertensive drugs T2DM: 15.6% Control: SBP < 140 and/or DBP < 90 mmHg T2DM: 10.3% | EH group: SBP: 141.0 ±14.0 DBP: 88.0±8.0 Control: SBP: 121.0 ±12.0 DBP: 77.0±7.0 | PCR-TaqMan | rs2228570 FokI $P_{HWE}$ = 0.52 rs11574129 $P_{HWE}$ >0.05 rs739837 $P_{HWE}$ >0.05 | Case = 2409 Male: 1137 (47.2%) Female: 1272 (52.8%) Control = 3063 Male: 1499 (48.9%) Female: 1564 (51.1%) | **FokI** EH group: FF: 472 (19.6%) Ff: 1180 (49.0%) ff: 756 (31.4%) Allele F: 2124 (44.1%) Allele f: 2692 (55.9%) Control: FF: 648 (21.2%) Ff: 1500 (49.1%) ff: 910 (29.8%) Allele F: 2796 (45.7%) Allele f: 3320 (54.3%) **rs1574129** EH group: TT: 1636 (67.9%) TC: 691 (28.7%) CC: 82 (3.4%) Allele T: 3963 (82.3%) Allele C: 855 (17.7%) Control: TT: 2098 (68.5%) TC: 864 (28.2%) CC: 99 (3.2%) Allele T: 5060 (82.7%) Allele C: 1062 (17.3%) **rs739837** EH group: TT: 1249 (52.0%) TC: 970 (40.3%) CC: 185 (7.7%) Allele T: 3468 (72.1%) Allele C: 1340 (27.9%) Control: TT: 1638 (53.5%) TC: 1186 (38.8%) CC: 235 (7.7%) Allele T: 4462 (72.9%) Allele C: 1656 (27.1%) |
| Cottone et al., 2015, Italy [53] | EH: 45.0 (IQR: 36–53.7) Control: N.R | EH group: all EH grade except malignant hypertension Control: Healthy | EH group: SBP: 140 (IQR: 131–155) DBP: 90 (IQR: 82–100) Control: N.R | PCR-RFLP | rs2228570 FokI $P_{HWE}$ = 0.38 rs1544410 BsmI $P_{HWE}$ = 0.62 | Case = 71 Male: 47 (66.2%) Female: 24 (33.8%) Control = 72 Male: N.R Female: N.R | **FokI** EH group: FF: 36 (50.7%) Ff: 30 (42.3%) ff: 5 (7.0%) Allele F: 102 (71.8%) Allele f: 40 (28.2%) Control: FF: 29 (40.3%) Ff: 36 (50.0%) ff: 7 (9.7%) Allele F: 94 (65.3%) Allele f: 50 (34.7%) **Bsm-I** EH group: BB: 16 (22.5%) Bb: 41 (57.8%) bb: 14 (19.7%) Allele B: 73 (51.4%) Allele b: 69 (48.6%) Control: BB: 15 (20.8%) Bb: 38 (52.8%) bb: 19 (26.4%) Allele B: 68 (47.2%) Allele b: 76 (52.8%) |
| Gussago et al., 2016, Italy [54] | EH: 102.3 ± 0.3 Control: 73.0 ± 0.6 | EH group: all EH > 100 years old Control: <100 years old | N.R | PCR-RFLP | rs2228570 FokI $P_{HWE}$ = 0.14 rs1544410 BsmI $P_{HWE}$ >0.05 rs7975232 ApaI $P_{HWE}$ >0.05 rs731236 TaqI $P_{HWE}$ >0.05 | Case = 102 Control = 163 | **FokI** EH group: FF: 45 (47.4%) Ff: 40 (42.1%) ff: 10 (10.5%) Allele B: 130 (68.4%) Allele b: 60 (31.6%) Control: FF: 79 (48.4%) Ff: 63 (38.7%) ff: 21 (12.9%) Allele B: 221 (67.8%) Allele b: 105 (32.2%) **Bsm-I** EH group: BB: 23 (25.8%) Bb: 44 (49.5%) bb: 22 (24.7%) Allele B: 90 (50.6%) Allele b: 88 (49.4%) Control: BB: 42 (25.8%) Bb: 56 (34.3%) bb: 65 (39.9%) Allele B: 140 (42.9%) Allele b: 186 (57.1%) **Apa-I** EH group: AA: 30 (31.9%) Aa: 57 (60.6%) aa: 7 (7.5%) Allele A: 117 (62.2%) Allele a: 71 (37.8%) Control: AA: 43 (26.7%) Aa: 85 (52.8%) aa: 33 (20.5%) Allele A: 171 (53.1%) Allele a: 151 (46.9%) **Taq-I** EH group: TT: 25 (31.7%) Tt: 43 (54.4%) tt: 11 (13.9%) Allele T: 93 (58.8%) Allele t: 65 (41.2%) Control: TT: 63 (40.9%) Tt: 77 (50.0%) tt: 14 (9.1%) Allele T: 203 (65.9%) Allele t: 105 (34.1%) |
| Lai et al., 2016, China [55] | EH: 61.0 ± 11.6 Control: 50.0 ± 14.9 | EH group: SBP ≥ 140 and/or DBP ≥ 90 mmHg Control: Healthy | EH group: SBP: 156.6 ± 16.8 DBP: 88.0 ± 11.7 Control: SBP: 121.4 ± 10.5 DBP: 73.6 ± 7.9 | PCR-RFLP | rs2228570 FokI $P_{HWE}$ = 0.15 | Case = 212 Male: 110 (51.9%) Female: 102 (48.1%) Control = 315 Male: 157 (49.8%) Female: 158 (50.2%) | **FokI** EH group: FF: 55 (25.9%) Ff: 116 (54.7%) ff: 41 (19.3%) Allele F: 226 (53.3%) Allele f: 198 (46.7%) Control: FF: 124 (39.4%) Ff: 137 (43.5%) ff: 54 (17.1%) Allele F: 385 (61.1%) Allele f: 245 (38.9%) |

(*Continued*)

**Table 1.** (Continued)

| Study, year, country | Baseline age (mean ± SD / SE / min - max) | Comorbidity | Baseline BP | Genotyping methods | SNP loci (p$_{HWE}$ for control group) | Sample Size | Results | | | |
|---|---|---|---|---|---|---|---|---|---|---|
| | | | | | | | **CYP2R1 rs1993116** | **CYP2R1 rs10741657** | **CYP24A1 rs4809957** | **CYP24A1 rs6068816** |
| Ye et al., 2018, China [56] | EH: 55.29 ± 8.68 Control: 54.52 ± 7.82 | EH group: SBP ≥ 140 and/or DBP ≥ 90 mmHg, or treated by antihypertensive drugs Control: Healthy | EH group: SBP: 146.47 ± 14.53 DBP: 90.24 ± 9.82 Control: SBP: 121.57 ± 10.19 DBP: 76.47 ± 7.42 | PCR-TaqMan | CYP2R1 rs1993116 Wild = CC Heterozygote = CT Recessive = TT rs10741657 Wild = CC Heterozygote = CT Recessive = TT P$_{HWE}$>0.05 CYP24A1 rs4809957 Wild = AA Heterozygote = AG Recessive = GG rs6068816 Wild = CC Heterozygote = CT Recessive = TT P$_{HWE}$>0.05 | Case = 394 Male: 281 (71.3%) Female: 131 (28.7%) Control = 525 Male: 353 (67.2%) Female: 172 (32.8%) | EH group: CC: 171 (43.4%) CT: 176 (44.7%) TT: 47 (11.9%) Allele C: 518 (65.7%) Allele T: 270 (34.3%) Control: CC: 188 (35.8%) CT: 256 (48.8%) TT: 81 (15.4%) Allele C: 632 (60.2%) Allele T: 418 (39.8%) | EH group: CC: 175 (44.4%) CT: 171 (43.4%) TT: 48 (12.2%) Allele C: 521 (66.1%) Allele T: 267 (33.9%) Control: CC: 196 (37.3%) CT: 249 (47.4%) TT: 80 (15.2%) Allele C: 641 (61.0%) Allele T: 409 (39.0%) | EH group: AA: 53 (13.5%) AG: 197 (50.0%) GG: 144 (36.5%) Allele C: 303 (38.5%) Allele T: 485 (61.5%) Control: CC: 82 (15.6%) CT: 244 (46.5%) TT: 199 (37.9%) Allele C: 408 (38.9%) Allele T: 642 (61.1%) | EH group: CC: 147 (37.3%) CT: 189 (48.0%) TT: 58 (14.7%) Allele C: 483 (61.3%) Allele T: 305 (38.7%) Control: CC: 219 (41.7%) CT: 225 (42.9%) TT: 81 (15.4%) Allele C: 663 (63.1%) Allele T: 387 (36.9%) |
| Zhang et al., 2018, China [57] | EH: 47.3 ± 15.7 Control: 46.5 ± 12.7 | EH group: SBP ≥ 140 and/or DBP ≥ 90 mmHg Control: Healthy | N.R | PCR-TaqMan | rs2228570 FokI P$_{HWE}$>0.05 rs7975232 ApaI P$_{HWE}$>0.05 | Case = 289 Male: 132 (45.7%) Female: 157 (54.3%) Control = 650 Male: 315 (48.5%) Female: 335 (51.5%) | **Fok-I** EH group: FF: 78 (27.1%) Ff: 160 (55.4%) ff: 51 (17.5%) Allele F: 316 (54.7%) Allele f: 262 (45.3%) Control: FF: 242 (37.2%) Ff: 300 (46.1%) ff: 108 (16.7%) Allele F: 784 (60.3%) Allele f: 516 (39.7%) | | **Apa-I** EH group: AA: 121 (41.8%) Aa: 138 (47.8%) aa: 30 (10.4%) Allele A: 380 (65.7%) Allele a: 198 (34.3%) Control: AA: 297 (45.7%) Aa: 275 (42.3%) aa: 78 (12.0%) Allele A: 869 (66.8%) Allele a: 431 (33.2%) | |
| Xia et al., 2019, China [41] | EH: 75.8 ± 10.6 Control: 72.9 ± 11.2 | EH group: SBP ≥ 140 and/or DBP ≥ 90 mmHg, or treated by antihypertensive drugs Control: Healthy | EH group: SBP: 131.6 ± 12.3 DBP: 71.6 ± 9.2 Control: SBP: 122.8 ± 9.8 DBP: 70.2 ± 7.9 | PCR-RFLP | rs2228570 FokI P$_{HWE}$>0.05 rs1544410 BsmI P$_{HWE}$>0.05 rs7975232 ApaI P$_{HWE}$>0.05 rs731236 TaqI P$_{HWE}$>0.05 | Case = 228 Male: 228 (100%) Female: 0 (0%) Control = 184 Male: 184 (100%) Female: 0 (0%) | **Fok-I** EH group: FF: 116 (50.9%) Ff: 98 (43.0%) ff: 14 (6.1%) Allele F: 330 (72.4%) Allele f: 126 (27.6%) Control: FF: 81 (44.0%) Ff: 81 (44.0%) ff: 22 (12.0%) Allele F: 243 (66.0%) Allele f: 125 (34.0%) | **Bsm-I** EH group: BB: 202 (88.6%) Bb: 22 (9.6%) bb: 4 (1.8%) Allele B: 426 (93.4%) Allele b: 30 (6.6%) Control: BB: 161 (87.5%) Bb: 23 (12.5%) bb: 0 (0%) Allele B: 345 (93.8%) Allele b: 23 (6.3%) | **Apa-I** EH group: AA: 23 (10.1%) Aa: 80 (35.1%) aa: 125 (54.8%) Allele A: 126 (27.6%) Allele a: 330 (72.4%) Control: AA: 10 (5.4%) Aa: 80 (43.5%) aa: 94 (51.1%) Allele A: 100 (27.2%) Allele a: 268 (72.8%) | **Taq-I** EH group: TT: 210 (92.1%) Tt: 18 (7.9%) tt: 0 (0%) Allele T: 438 (96.1%) Allele t: 18 (3.9%) Control: TT: 169 (91.8%) Tt: 15 (8.2%) tt: 0 (0%) Allele T: 353 (95.9%) Allele t: 15 (4.1%) |
| Obukhova et al., 2020, Ukraine [42] | EH: 64.7 ± 0.73 Control: 76.7 ± 0.93 | EH group: EH with 63.7% having IAS Control: Normal blood pressure with 46.7% having IAS | N.R | PCR-RFLP | rs2228570 FokI P$_{HWE}$>0.05 rs1544410 BsmI P$_{HWE}$>0.05 rs7975232 ApaI P$_{HWE}$>0.05 rs731236 TaqI P$_{HWE}$>0.05 | Case = 201 Control = 90 | **Fok-I** EH group: FF: 51 (25.4%) Ff: 107 (53.2%) ff: 43 (21.4%) Allele F: 209 (52.0%) Allele f: 193 (48.0%) Control: FF: 22 (24.4%) Ff: 42 (46.7%) ff: 26 (28.9%) Allele F: 86 (47.8%) Allele f: 94 (52.2%) | **Bsm-I** EH group: BB: 26 (12.9%) Bb: 87 (43.3%) bb: 88 (43.8%) Allele B: 139 (34.6%) Allele b: 263 (65.4%) Control: BB: 13 (14.4%) Bb: 38 (42.2%) bb: 39 (43.3%) Allele B: 64 (35.6%) Allele b: 116 (64.4%) | **Apa-I** EH group: AA: 49 (24.4%) Aa: 92 (45.8%) aa: 60 (29.9%) Allele A: 190 (47.3%) Allele a: 212 (52.3%) Control: AA: 23 (25.6%) Aa: 43 (47.8%) aa: 24 (26.7%) Allele A: 89 (49.4%) Allele a: 91 (50.6%) | **Taq-I** EH group: TT: 85 (42.3%) Tt: 93 (46.3%) tt: 23 (11.4%) Allele T: 263 (65.4%) Allele t: 139 (34.6%) Control: TT: 36 (40.0%) Tt: 43 (47.8%) tt: 11 (12.2%) Allele T: 115 (63.9%) Allele t: 65 (36.1%) |

*(Continued)*

**Table 1.** (Continued)

| Study, year, country | Baseline age (mean ± SD / SE / min - max) | Comorbidity | Baseline BP | Genotyping methods | SNP loci (P$_{HWE}$ for control group) | Sample Size | Results |
|---|---|---|---|---|---|---|---|
| Varakantham et al., 2020, India [43] | EH: 53.0 ± 13.01 Control: 43.1 ± 12.58 | EH group: SBP ≥ 140 and/or DBP ≥ 90 mmHg, or treated by antihypertensive drugs Control: Healthy | EH group: SBP: 140.5 ± 16.68 DBP: 90.4 ± 11.76 Control: SBP: 120.1 ± 15.22 DBP: 79.9 ± 9.9 | PCR–TaqMan | CYP24A1 rs2762939 Wild = GG Heterozygote = GC Recessive = CC P$_{HWE}$>0.05 | Case = 292 Male: 132 (45.2%) Female: 160 (54.8%) Control = 324 Male: 158 (48.8%) Female: 166 (51.2%) | **CYP24A1 rs2762939** EH group: GG: 138 (47.3%) GC: 126 (43.2%) CC: 28 (9.6%) Allele G: 402 (68.8%) Allele C: 182 (31.2%) Control: GG: 148 (45.7%) GC: 144 (44.4%) CC: 32 (9.9%) Allele G: 440 (67.9%) Allele C: 208 (32.1%) |
| Prasad et al., 2021, India [44] | Between 25–60 years old | EH group: SBP ≥ 140 and/or DBP ≥ 90 mmHg Control: Healthy | EH group: SBP: 134.53 ± 6.78 DBP: 89.32 ± 3.00 Control: SBP: 116.65 ± 4.13 DBP: 76.97 ± 4.69 | PCR–RFLP | rs2228570 FokI P$_{HWE}$ = 0.64 | Case = 200 Male: 84 (42%) Female: 116 (58%) Control = 200 Male: 103 (51.5%) Female: 97 (48.5%) | **Fok-I** EH group: FF: 131 (65.5%) Ff: 6 (3%) ff: 63 (31.5%) Allele F: 268 (67%) Allele f: 132 (33%) Control: FF: 134 (67%) Ff: 58 (29%) ff: 8 (4%) Allele F: 326 (82%) Allele f: 74 (18%) |
| Repchuk et al., 2021, Ukraine [45] | EH: 59.86 ± 6.22 Control: N.R | EH group: hypertension-mediated organ damage (target organs damage–2nd severity stage, asymptomatic disease), grade 1–3; moderate-high CV risk, dyslipidemia Control: Healthy | N.R | PCR–TaqMan | rs2228570 FokI P$_{HWE}$>0.05 | Case = 100 Male: 21 (21%) Female: 79 (79%) Control = 60 Male: N.R Female: N.R | **Fok-I** EH group: FF: 27 (27%) Ff: 50 (50%) ff: 23 (23%) Allele F: 104 (52.0%) Allele f: 96 (48.0%) Control: FF: 14 (23.3%) Ff: 28 (46,7%) ff: 18 (30.0%) Allele F: 56 (46.7%) Allele f: 64 (53.3%) |
| Zehra et al., 2022, Pakistan [46] | Stage I: 52 ± 11.60 Stage II: 52.48 ± 12.72 Stage III: 52.85 ± 13.47 Control: 51.45 ± 09.78 | EH group: Hypertensive coronary heart disease Stage I: SBP 140–159 mmHg and DBP 90–99 mmHg Stage II: SBP 160–179 mmHg and DBP 100–109 mmHg Stage III: SBP ≥ 180 mmHg and/or DBP ≥ 110 mmHg Control: Healthy | Stage I: SBP: 142.15 ± 4.12 DBP: 93.62 ± 8.94 Stage II: SBP: 163.40 ± 4.78 DBP: 107.87 ± 12.14 Stage III: SBP: 194.81 ± 17.62 DBP: 123.29 ± 19.98 Control: SBP: 121.7 ± 0.7 DBP: 75.5 ± 1.06 | PCR–ARMS | rs731236 TaqI P$_{HWE}$>0.05 | EH group: Stage I: 176 Stage II: 47 Stage III: 27 Control: 250 | **Taq-I** EH group: TT: 26 (10.4%) Tt: 191 (76.4%) tt: 33 (13.2%) Allele T: 243 (48.6%) Allele t: 257 (51.4%) Control: TT: 102 (40.8%) Tt: 128 (51.2%) tt: 20 (8%) Allele T: 332 (66.4%) Allele t: 168 (33.6%) |

(*Continued*)

**Table 1.** (Continued)

| Study, year, country | Baseline age (mean ± SD / SE / min - max) | Comorbidity | Baseline BP | Genotyping methods | SNP loci ($P_{HWE}$ for control group) | Sample Size | Results | | | |
|---|---|---|---|---|---|---|---|---|---|---|
| | | | | | | | **Fok-I** | **Bsm-I** | **Apa-I** | **Taq-I** |
| Gariballa et al., 2023, United Arab Emirates [47] | All group: 41 ± 12 | EH group: SBP ≥ 140 and/or DBP ≥ 90 mmHg Control: Normal blood pressure | N.R | PCR-TaqMan | rs2228570 FokI $P_{HWE}$>0.05 rs1544410 BsmI $P_{HWE}$>0.05 rs731236 TaqI $P_{HWE}$>0.05 rs7975232 ApaI $P_{HWE}$>0.05 | Case: 41 Control: 236 | EH group: FF: 16 (39%) Ff: 14 (34%) ff: 11 (27%) Allele F: 46 (56.1%) Allele f: 36 (43.9%) Control: FF: 102 (43%) Ff: 112(47%) ff: 22 (9%) Allele F: 316 (66.9%) Allele f: 156 (33.1%) | EH group: BB:12 (29%) Bb: 20 (49%) bb: 9 (22%) Allele B: 44 (53.7%) Allele b: 38 (46.3%) Control: BB: 93 (39%) Bb: 95 (40%) bb: 48 (20%) Allele B: 281 (59.5%) Allele b: 191 (40.5%) | EH group: AA:20(49%) Aa: 16 (39%) aa: 5 (12%) Allele A: 56 (68.3%) Allele a: 26 (31.7%) Control: AA: 105 (44%) Aa 90 (38%) aa: 41 (17%) Allele A: 300 (63.6%) Allele a: 172 (36.4%) | EH group: TT: 9 (22%) Tt: 24 (59%) tt: 8 (20%) Allele T: 42 (51.2%) Allele t: 40 (48.8%) Control: TT: 78 (33%) Tt: 113 (48%) tt: 45 (19%) Allele T: 269 (57.0%) Allele t: 203 (43.0%) |
| Rojo-Tolosa et al., 2023, Spain [48] | EH: 66 (60–73) Control: 65 (60–73) | EH group: SBP ≥ 140 and/or DBP ≥ 90 mmHg, or treated by antihypertensive drugs Obesity: 44.5% Dyslipidemia: 49.2% Diabetes: 41.6% Control: Normal blood pressure Obesity: 27.4% Dyslipidemia: 26.6% Diabetes: 10.9% | EH group: SBP: 137 (121–154) DBP: 80 (70–88) Control: N.R | PCR-TaqMan | rs2228570 FokI $P_{HWE}$>0.05 rs1544410 BsmI $P_{HWE}$>0.05 rs731236 TaqI $P_{HWE}$>0.05 rs7975232 ApaI $P_{HWE}$>0.05 | Case = 250 Male: 119 (47.6%) Female: 131 (52.4%) Control = 500 Male: 233 (46.6%) Female: 267 (53.4%) | EH group: FF: 105 (43.8%) Ff: 99 (41.2%) ff: 36 (15%) Allele F: 309 (64.4%) Allele f: 171 (35.6%) Control: FF: 212 (42.7%) Ff: 212 (42.7%) ff: 72 (14.6%) Allele F: 636 (64.1%) Allele f: 356 (35.9%) | EH group: BB: 88 (36.2%) Bb: 106 (43.6%) bb: 49 (20.2%) Allele B: 282 (58.0%) Allele b: 204 (42.0%) Control: BB: 177 (36.0%) Bb: 220 (44.8%) bb: 94 (19.1%) Allele B: 574 (58.5%) Allele b: 408 (41.5%) | EH group: AA:76 (31.5%) Aa: 112 (46.5%) aa: 53 (22%) Allele B: 264 (54.8%) Allele a: 218 (45.2%) Control: AA: 141 (29.1%) Aa: 228 (47.1%) aa: 115 (23.8%) Allele A: 510 (53.2%) Allele a: 458 (46.8%) | EH group: TT: 88 (35.3%) Tt: 116 (46.6%) tt: 45 (18.1%) Allele T: 292 (58.6%) Allele t: 206 (41.4%) Control: TT: 177 (37.6%) Tt: 215 (45.6%) tt: 79 (16.8%) Allele T: 569 (60.4%) Allele t: 373 (39.6%) |
| Nabil et al., 2024, Bangladesh [49] | EH: 39.48 ± 1.24 Control: 37.80 ± 1.79 | EH group: SBP ≥ 140 and/or DBP ≥ 90 mmHg, or treated by antihypertensive drugs Control: Healthy | EH group: SBP: 141.2 ± 16.0 DBP: 90.5 ± 9.72 Control: SBP: 109.6 ± 7.09 DBP: 71.07 ± 6.71 | PCR-RFLP | rs2228570 FokI $P_{HWE}$>0.05 rs1544410 BsmI $P_{HWE}$>0.05 rs731236 TaqI $P_{HWE}$>0.05 rs7975232 ApaI $P_{HWE}$>0.05 | Case = 111 Male: 78 (70.4%) Female: 33 (29.6%) Control = 100 Male: 68 (68.5%) Female: 32 (31.5%) | EH group: FF: 11 (9.7%) Ff: 39 (34.2%) ff: 64 (56.1%) Allele F: 61 (26.8%) Allele f: 167 (73.2%) Control: FF: 9 (9%) Ff: 56 (56%) ff: 35 (35%) Allele F: 74 (37%) Allele f: 126 (63%) | EH group: BB: 11 (9.9%) Bb: 64 (57.7%) bb: 36 (32.4%) Allele B: 86 (38.7%) Allele b: 136 (61.3%) Control: BB: 10 (10.0%) Bb: 58 (58.0%) bb: 32 (32.0%) Allele B: 78 (39%) Allele b: 122 (61%) | | EH group: TT: 51 (46.0%) Tt: 44 (39.6%) tt: 16 (14.4%) Allele T: 146 (65.8%) Allele t: 76 (34.2%) Control: TT: 44 (44.0%) Tt: 49 (49.0%) tt: 7 (7.0%) Allele T: 137 (68.5%) Allele t: 63 (31.5%) |
| **CROSS SECTIONAL STUDY** | | | | | | | | | | |
| Vaidya et al., 2011, United States [58] | EH: 48.3 ± 8.2 Control: 39.9 ± 10.9 | EH group: untreated seated DBP >100 mmHg, a DBP >90 mmHg with one or more antihypertensive medication Control: normotensive with 3 consecutive seated blood pressure <140/90 mmHg, and no first-degree relatives diagnosed | EH group: Low dietary sodium balance SBP: 132.2 ± 18.4 DBP: 79.9 ± 11.0 High dietary sodium balance SBP: 146.8 ± 20.3 DBP: 87.5 ± 11.4 | PCR-TaqMan | rs2228570 FokI $P_{HWE}$ = 0.95 | Case = 246 Male: 114 (46.3%) Female: 132 (53.7%) Control = 246 Male: 104 (42.3%) Female: 142 (57.7%) | EH group: FF: 119 (31.7%) Ff: 195 (52%) ff: 61 (16.3%) Allele F: 433 (58.0%) Allele f: 317 (42.0%) Control: FF: 62 (42.5%) | | | |

**Table 1.** (Continued)

| Study, year, country | Baseline age (mean ± SD / SE / min - max) | Comorbidity | Baseline BP | Genotyping methods | SNP loci ($P_{HWE}$ for control group) | Sample Size | Results |
|---|---|---|---|---|---|---|---|
| | | with hypertension prior to the age of 60 years | Control: Low dietary sodium balance SBP: 105.3 ± 10.3 DBP: 63.1 ± 7.0 High dietary sodium balance SBP: 109.8 ± 11.4 DBP: 66.1 ± 8.1 | | | | Ff: 66 (45.2%) ff: 18 (12.3%) Allele F: 190 (65.0%) Allele f: 102 (35.0%) |
| Santos et al., 2021, Brazil [59] | EH: 130–139 and/or 80–89 mmHg (Stage 1): 55.05 ± 5.10 ≥140 and/or ≥90; Stage 2 (Stage 2): 57.85 ± 5.37 Control: <120 and <80 mmHg (Normal): 54.00 ± 5.11 120–129 and <80 mmHg (Elevated): 57.58 ± 6.01 | Case: SBP ≥ 140 and/or DBP ≥ 90 mmHg Obesity: 35.2% Metabolic syndrome: 50.4% Control: SBP < 140 and/or DBP < 90 mmHg Other comorbidity; Normotension: Obesity: 8.3% Metabolic syndrome: 11.7% Elevated: Obesity: 26.9% Metabolic syndrome: 15.4% Stage 1: Obesity: 21.6% Metabolic syndrome: 25.8% | Case: Stage 2: SBP: 144.91 ± 16.53 DBP: 91.00 ± 9.46 Control: Normotension: SBP: 107.19 ± 6.14 DBP: 67.78 ± 4.78 Elevated: SBP: 121.00 ± 2.21 DBP: 70.39 ± 4.01 Stage 1: SBP: 122.32 ± 7.82 DBP: 79.80 ± 3.11 | PCR-TaqMan | rs1544410 BsmI $P_{HWE}$ >0.05 rs731236 TaqI $P_{HWE}$ >0.05 | Case = ≥140 and/or ≥90; Stage 2 (Stage 2): 127 Control: <120 and <80 mmHg (Normal): 60 120–129 and <80 mmHg (Elevated): 26 130–139 and/or 80–89 mmHg (Stage 1): 123 | **Bsm-I** Case: BB + Bb: 109 (85.8%) bb: 18 (14.2%) Control: BB + Bb: 159 (76.8%) bb: 48 (23.2%) **Taq-I** Case: TT + Tt: 116 (91.3%) tt: 11 (8.7%) Control: TT + Tt: 171 (82.2%) tt: 37 (17.8%) |
| COHORT STUDY | | | | | | | |
| Wang et al., 2013, United States [60] | EH: 57.1 ± 7.5 Control: 57.5 ± 8.2 | EH group: SBP ≥ 140 and/or DBP ≥ 90 mmHg, or treated by antihypertensive drugs Control: Healthy | EH group: SBP: 123.9 ± 8.0 DBP: 77.7 ± 5.8 Control: SBP: 119.0 ± 8.5 DBP: 74.7 ± 6.3 | PCR-RFLP | rs2228570 FokI $P_{HWE}$ = 0.34 rs1544410 BsmI $P_{HWE}$ = 0.12 | Case = 537 Control = 376 | **Fok-I** EH group: FF: 208 (39.8%) Ff: 229 (43.9%) ff: 85 (16.3%) Allele F: 645 (61.8%) Allele f: 399 (38.2%) Control: FF: 139 (40.1%) Ff: 178 (43.7%) ff: 46 (35.1%) Allele F: 456 (62.8%) Allele f: 270 (36.2%) **Bsm-I** EH group: BB: 80 (60.6%) Bb: 265 (63.1%) bb: 182 (52.6%) Allele B: 425 (40.3%) Allele b: 629 (59.7%) Control: BB: 52 (39.4%) Bb: 155 (36.9%) bb: 164 (47.4%) Allele B: 259 (34.9%) Allele b: 483 (65.1%) |

ARMS: Amplification Refractory Mutation System; DBP: diastolic blood pressure; EH: essential hypertension; HWE: Hardy-Weinberg Equilibrium; IAS: ischemic acute stroke; PCR-RFLP: Polymerase Chain Reaction - Restriction Fragment Length Polymorphism; SBP: systolic blood pressure; SD: Standard deviation

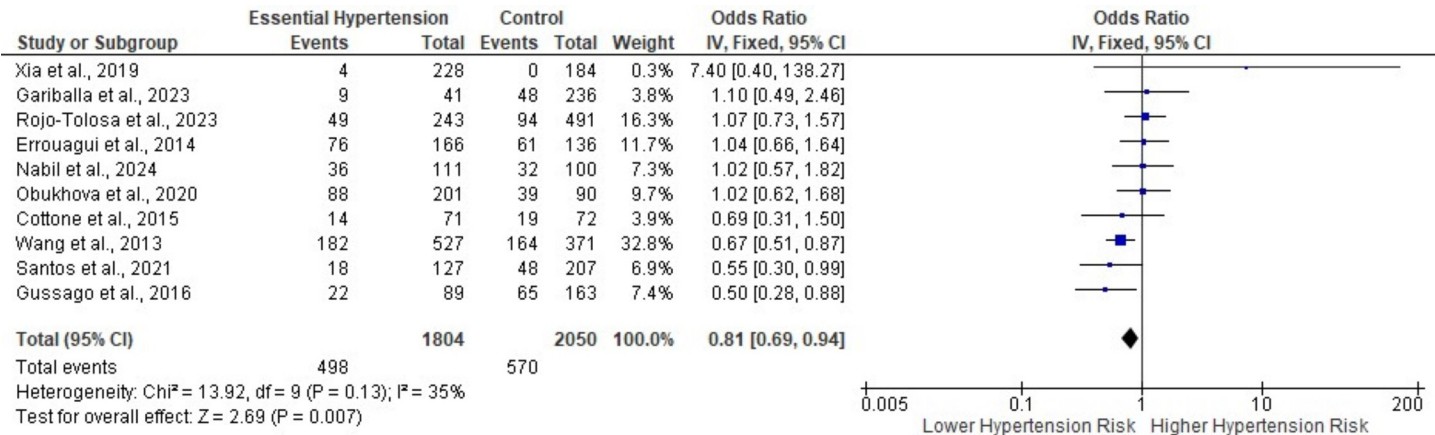

**Fig 2. Forest plot for recessive model (Genotype bb vs Genotype BB + Bb) of BsmI polymorphism in risk of hypertension.** The solid squares denote the odds ratios, with the horizontal lines indicating the 95% confidence intervals and the diamond denotes the pooled effect size. SD, standard deviation; IV, inverse variance; CI, confidence interval; df, degrees of freedom; Chi2, chi-squared statistic; p, p-value; I2, I-squared heterogeneity statistic; Z, Z statistic.

## Risk of essential hypertension in patients with ApaI polymorphism

Seven studies investigated the susceptibility to EH associated with the ApaI polymorphism (S18–S21 Figs) [41, 42, 47, 48, 51, 54, 57]. The analysis showed no significant risk of EH for patients with the ApaI polymorphism across the allele model (OR: 1.04; 95%CI, 0.94 to 1.16; p = 0.45), additive model (OR: 0.83; 95%CI, 0.66 to 1.05; p = 0.13), dominant model (OR: 0.97; 95%CI, 0.82 to 1.15; p = 0.75), and recessive model (OR: 0.93; 95%CI, 0.77 to 1.12; p = 0.46). Due to the absence of significant statistical heterogeneity, the FEM was applied for all analyses. Subgroup analyses of all models based on continent of origin also yielded no significant results (S22–S25 Figs). Sensitivity analysis using a leave-one-out approach for the all models demonstrated no changes in the pooled results (S26 Fig).

## Risk of essential hypertension in patients with TaqI polymorphism

Five studies evaluated the association between the TaqI polymorphism and EH susceptibility (S27–S30 Figs) [40, 42, 46, 54]. Additionally, one study reported only on the recessive model [59]. The analysis indicated that the TaqI polymorphism was not associated with EH risk in the allele model (OR: 0.90; 95%CI, 0.67 to 1.21; p = 0.48), additive model (OR: 1.36; 95%CI, 0.65 to 2.82; p = 0.41), dominant model (OR: 1.31; 95%CI, 0.78 to 2.21; p = 0.31), and recessive model (OR: 0.99; 95%CI, 0.63 to 1.54; p = 0.95). Due to significant statistical heterogeneity (p < 0.1), a REM was employed. Subgroup analyses by continent of origin for all models showed no significant findings, except for the recessive models in the American subgroup, which were significant but based on only one study (S31–S34 Figs) [59]. Sensitivity analysis using a leave-one-out approach for all models demonstrated no changes in the pooled results (S35 Fig).

## Risk of essential hypertension in patients with other single nucleotide polymorphisms

Other SNPs were reported in only two studies [43, 56]. Ye et al. [56] examined the role of CYP2R1 rs1993116 and rs10741657, and CYP24A1 rs4809957 and rs6068816 in EH susceptibility. They found a significantly lower SBP in treated EH patients with the TT genotype of rs1993116 and rs10741657 (p = 0.002). For adjusted additive and dominant models, there was a

significant reduction in EH risk associated with rs1993116, resulting in an OR of 0.788 (95%CI, 0.644–0.963; p = 0.02) and 0.719 (95%CI, 0.545–0.949; p = 0.02), respectively. The results for rs10741657 were also significant, with an OR of 0.805 (95% CI, 0.66–0.983; p = 0.033) for the additive model and 0.733 (95% CI, 0.556–0.966; p = 0.028) for the dominant model. There were no significant associations between CYP24A1 rs4809957 and rs6068816 with EH susceptibility.

Meanwhile, Varakantham et al. reported that patients with the CYP24A1 rs2762939 CC variant had a significantly higher risk of EH in males (OR: 3.141; 95% CI, 1.164 to 8.478; p = 0.024) and an inverse association in females (OR: 0.398; 95% CI, 0.172 to 0.920; p = 0.031) [43].

The complete summary for the risk of essential hypertension across different polymorphisms (FokI, BsmI, ApaI, and TaqI) and genetic models are available in Table 2.

## Additional analysis

We conducted an additional meta-analysis that included only case-control studies. The pooled analysis from 13 case-control studies [39–57] showed no significant association between the FokI polymorphism and EH risk across all models (S36 Fig). For the BsmI polymorphism, the pooled analysis became non-significant for all models after restricting to case-control studies (S37 Fig). The analysis for the ApaI polymorphism also indicated no significant risk of EH across all models (S38 Fig). Similarly, the pooled analysis for the TaqI polymorphism revealed no significant association with EH risk after the inclusion of case-control studies only (S39 Fig).

## Meta regression

Meta-regression analysis revealed that methodological quality significantly influenced the pooled effect for EH risk in patients with the FokI polymorphism based on the allele, additive, and dominant models, but not the recessive model (all p < 0.0005) (S40–S43 Figs). The proportion of male participants significantly influenced the pooled effect for EH risk in the additive model for the FokI polymorphism (p = 0.0235) (S41 Fig). Age significantly influenced the pooled effect for EH risk in the recessive model for the FokI polymorphism (p = 0.0327) (S43 Fig).

**Table 2. Summary of essential hypertension risk in patients with FokI, BsmI, ApaI, and TaqI polymorphism.**

| Polymorphism | Genetic Model | OR (95% CI) | p-value | Heterogeneity ($I^2$) |
|---|---|---|---|---|
| FokI | Allele model | 0.96 (0.84–1.10) | 0.55 | 78% |
|  | Additive model | 1.08 (0.82–1.44) | 0.58 | 77% |
|  | Dominant model | 1.00 (0.84–1.18) | 0.99 | 69% |
|  | Recessive model | 1.11 (0.86–1.44) | 0.42 | 79% |
| BsmI | Allele model | 1.08 (0.97–1.20) | 0.17 | 0% |
|  | Additive model | 0.91 (0.73–1.14) | 0.42 | 0% |
|  | Dominant model | 0.98 (0.81–1.18) | 0.81 | 0% |
|  | Recessive model | 0.81 (0.69–0.94) | 0.007 | 35% |
| ApaI | Allele model | 1.04 (0.94–1.16) | 0.45 | 0% |
|  | Additive model | 0.83 (0.66–1.05) | 0.13 | 17% |
|  | Dominant model | 0.97 (0.82–1.15) | 0.75 | 0% |
|  | Recessive model | 0.93 (0.77–1.12) | 0.46 | 32% |
| TaqI | Allele model | 0.90 (0.67–1.21) | 0.48 | 81% |
|  | Additive model | 1.36 (0.65–2.82) | 0.41 | 82% |
|  | Dominant model | 1.31 (0.78–2.21) | 0.31 | 87% |
|  | Recessive model | 0.99 (0.63–1.54) | 0.95 | 66% |

Meta-regression analysis for the BsmI polymorphism did not reveal any significant influence of NOS score and age on the EH risk based on the recessive model (p = 0.3463 and p = 0.1575, respectively) (S44 Fig). Meta-regression for other models of the BsmI polymorphism was not conducted due to insufficient information. Similarly, meta-regression for other polymorphisms was not generated due to inadequate data.

For the FokI polymorphism, studies with higher NOS scores tended to show a lower risk of EH for the F allele and a higher risk of EH for recessive homozygous individuals (ff). Studies with a higher proportion of male participants tended to show a lower risk of EH for recessive homozygous individuals (ff), indicating that studies with a higher proportion of female participants tended to show a higher risk of EH for recessive homozygous individuals (ff). Additionally, studies with older patient populations tended to show a lower risk of EH for recessive homozygous individuals (ff), whereas studies with younger patients tended to show a higher risk of EH for recessive homozygous individuals (ff).

## Quality assessment and publication bias

Quality assessment using NOS showed that all case-control studies' results range from 4 to 8. There were 5 studies scored with 4 out of 9 [39, 40, 44, 45, 53], 4 studies scored with 5 out of 9 [42, 46, 50, 51], 2 studies scored with 6 out of 9 [43, 54], and 3 studies scored 7 out of 9 [47, 52, 56], and 5 studies scored 8 out of 9 [41, 48, 49, 55, 57]. For cross-sectional studies, 1 study [58] scored 10 out of 10 while another study [59] scored 9 out of 10. A cohort study scored 8 out of 9 [60]. Complete reasoning can be found in S2–S4 Tables.

A funnel plot with trim and fill analysis is generated (S45–S49 Figs). Egger's test showed no publication bias for all models (All p > 0.05). Rank correlation test also showed no publication bias for all models (All p > 0.05).

## Discussions

In this review, we examined the association between various VDR polymorphisms and EH risk. No significant association was found between the FokI polymorphism and EH risk, a result that remained unchanged after sensitivity analysis, suggesting no study had a disproportionate influence on the overall effect-size estimate. Furthermore, considering that ethnicity and age are important factors in studying EH, as indicated by previous research, we conducted additional related analyses [61]. The subgroup analysis indicated that continent of origin did not influence the results. However, meta-regression revealed that age and sex significantly affected the findings. The impact of the FokI polymorphism on EH risk varied according to sex, with the recessive homozygous (ff) genotype associated with a higher risk of EH in females but a lower risk in males. Additionally, younger individuals with the ff genotype had a higher EH risk, while older individuals had a lower risk. Methodological quality, assessed by NOS score, also significantly influenced EH risk, with higher quality studies indicating a higher risk for those with the ff genotype. Meanwhile, our meta-analysis of the BsmI polymorphism indicated that, in the recessive model analysis, individuals with the recessive homozygous (bb) genotype exhibited a lower risk of EH. This polymorphism involves a nucleotide substitution (A to G in intron 8) that affects transcript stability, reducing VDR protein levels and potentially increasing RAS activity and blood pressure [61].

Three of the four single nucleotide polymorphisms (SNPs) occur within intron sections (the TaqI, ApaI, and BsmI variants), whereas the FokI variant is the only one that alters the codon. The BsmI and TaqI polymorphisms do not alter the structure of the VDR protein; however, they can affect the stability and/or translation efficiency of RNA [62]. Previous studies have reported associations between VDR gene polymorphisms and plasma renin activity [63].

Consequently, the influence of these VDR polymorphisms on EH risk has been investigated and confirmed by previous meta-analyses [21, 24]. According to Nunes et al. [24], the recessive homozygous (ff) genotype of the FokI polymorphism was associated with a lower risk of EH in the global population analysis. This effect was also evident in the Asian subgroups. However, their meta-analysis was based on small studies, comprising only six pooled studies, with evidence for the Asian subgroups derived from only two studies. Other polymorphisms were not examined in that review [24]. Meanwhile, another meta-analysis conducted by Zhu et al. [21] reported differing results, indicating no significant association between the FokI polymorphism and EH risk. However, they found that the BsmI polymorphism might increase EH risk. Their analysis showed that individuals with the homozygous dominant (BB) genotype, corresponding to the GG genotype, had a higher risk of EH compared to those with the homozygous recessive (bb) genotype, corresponding to the AA genotype [21]. Therefore, the substitution from A to G may indeed increase EH risk, a finding that is consistent with our meta-analysis.

The previous two meta-analyses did not undertake a comprehensive evaluation of the genetic interplay involving age, sex, and population origin in relation to EH risk. In our study, we found that age and sex significantly influence the association between the FokI polymorphism and EH risk. These findings may elucidate the existing discrepancies and inconsistent results concerning the FokI polymorphism's effect on EH risk. Therefore, it is crucial that future studies take age and sex into account when assessing the impact of the FokI polymorphism on EH risk. Conversely, our analysis of the BsmI polymorphism reveals a more consistent association with EH risk, which appears to be independent of age.

In parallel to the BsmI polymorphism, the Taq-I polymorphism, situated within exon 9 but with both exerting influence on the 3' untranslated region (3'UTR) of the VDR gene, assumes a pivotal role in modulating VDR gene expression. Specifically, it operates by regulating mRNA stability and protein translation efficiency [19]. The 3'UTR constitutes a locus for epigenetic phenomena, notably CpG island methylation, which demonstrates susceptibility to alterations contingent upon racial factors and the presence of polymorphisms within this gene region [20]. While these genetic variants do not induce alterations in the amino acid sequence of the resultant protein, they possess the capacity to modify the sequence within this regulatory segment of the gene, thereby exerting influence on gene expression and subsequent protein synthesis. Empirical evidence suggests that Bsm-I, Apa-I, and Taq-I exhibit linkage disequilibrium, with particularly robust linkage observed between Bsm-I and Taq-I [59]. This phenomenon implies that over time, restricted recombination within a specific gene locus fosters the emergence of polymorphisms intricately associated with one another, thereby potentially elucidating their concomitant effects across diverse biological processes [12]. Consequently, the linkage disequilibrium observed between the Bsm-I and Taq-I polymorphisms may be construed as a genetic determinant contributing, albeit partially, to the observed association of both variants with heightened blood pressure levels and susceptibility to EH [59]. In contrast to the available evidence, however, this meta-analysis did not uncover any significant association between the ApaI and TaqI polymorphisms and EH risk. This discrepancy could be attributed to the noted partial contribution of both polymorphisms towards EH susceptibility compared to the primary determinant, namely the BsmI polymorphism.

The differential association between VDR polymorphisms and EH risk could be attributed to the distinct biological functions of each polymorphism. For example, the BsmI polymorphism involves a nucleotide substitution in intron 8 that influences transcript stability, potentially reducing VDR protein expression [18]. This reduction may lead to increased RAS activity, contributing to elevated blood pressure and hypertension risk [19, 20]. Conversely, other polymorphisms like TaqI and ApaI, while located in similar non-coding regions, may not exert as pronounced an effect on gene expression. Studies have shown that TaqI and ApaI

polymorphisms, though associated with VDR mRNA stability, have less influence on downstream pathways that regulate blood pressure compared to BsmI [47, 54]. Moreover, the linkage disequilibrium observed between BsmI and TaqI suggests that these polymorphisms may exert combined effects, yet BsmI appears to play a more dominant role in modulating the VDR's influence on hypertension. This could explain why BsmI shows a stronger and more consistent association with EH risk in our analysis, while TaqI and ApaI do not. Additionally, FokI is unique in that it directly alters the structure of the VDR protein, leading to functional changes in receptor activity, but its impact on EH risk appears to be more dependent on population characteristics such as age and sex, as demonstrated in our meta-regression analysis.

Our study builds upon prior reviews, such as those conducted by Zhu et al. [21] and Nunes et al. [24], which examined the relationship between VDR polymorphisms and EH. We included additional studies published since these reviews and employed updated meta-regression techniques to assess the impact of age, sex, and study quality. This comprehensive approach allows for a more accurate understanding of the associations between VDR polymorphisms and EH risk.

Our meta-analysis encountered several limitations. Significant heterogeneity was observed in the results pertaining to the FokI and TaqI polymorphisms, likely stemming from variations across studies rather than sampling errors. The primary sources of heterogeneity were methodological quality, age, and sex differences among the studies, as indicated by meta-regression analysis. Despite conducting additional analyses by exclusively including studies with the same design, the heterogeneity persisted. Furthermore, potential publication bias may have affected our results, as smaller studies with null or negative findings are less likely to be published, which could have inflated the associations observed for certain polymorphisms. Although we used funnel plots and Egger's test to assess publication bias, the limited number of studies included in some analyses might have hindered our ability to detect bias accurately.

Another limitation arises from the variability in study designs, including differences in population demographics, genotyping methods, and diagnostic criteria for essential hypertension. This variability could have contributed to the inconsistent findings across studies, particularly for the ApaI and TaqI polymorphisms. Additionally, the scarcity of data prevented us from fully assessing the impact of sex on the BsmI polymorphism. Similarly, we were unable to investigate the influence of methodological quality, sex, and age on the ApaI and TaqI polymorphisms due to the limited number of available studies.

Further research is needed to explore the role of other environmental and genetic factors that may interact with VDR polymorphisms to influence EH risk. For instance, vitamin D levels, sun exposure, dietary intake, and lifestyle factors such as physical activity and smoking could modify the effects of these polymorphisms on hypertension [64]. Additionally, epigenetic mechanisms, including DNA methylation and histone modifications, may further influence how VDR gene expression affects blood pressure regulation [65]. Future studies should also consider investigating the combined impact of other polymorphisms, such as those in the RAS, on EH risk. Larger multi-ethnic cohort studies with standardized genotyping methods will be essential for validating these associations and identifying potential gene-environment interactions that may contribute to the development of hypertension. Such studies will help elucidate the broader genetic landscape underlying EH and improve personalized approaches to prevention and treatment.

## Conclusions

FokI polymorphism's impact on EH risk is influenced by sex, age, and study quality, with the recessive homozygous (ff) genotype associated with lower EH risk in males and older

individuals. Conversely, BsmI polymorphism is independently associated with EH risk, with recessive homozygotes having lower risk. No significant association was found between ApaI and TaqI polymorphisms and EH risk.

## Supporting information

**S1 Table. Keywords used in database searching.**
(DOCX)

**S2 Table. Newcastle Ottawa Scale for case-control studies.**
(DOCX)

**S3 Table. Newcastle Ottawa Scale for cross-sectional studies.**
(DOCX)

**S4 Table. Newcastle Ottawa Scale for cohort studies.**
(DOCX)

**S1 Fig. Forest plot for allele model (Allele F vs Allele f) of FokI polymorphism in risk of hypertension.**
(JPG)

**S2 Fig. Forest plot for additive model (Genotype ff vs Genotype FF) of FokI polymorphism in risk of hypertsension.**
(JPG)

**S3 Fig. Forest plot for dominant model (Genotype ff + Ff vs Genotype FF) of FokI polymorphism in risk of hypertension.**
(JPG)

**S4 Fig. Forest plot for recessive model (Genotype ff vs Genotype FF + Ff) of FokI polymorphism in risk of hypertension.**
(JPG)

**S5 Fig. Subgroup analysis according to continent for allele model (Allele F vs Allele f) of FokI polymorphism in risk of hypertension.**
(JPG)

**S6 Fig. Subgroup analysis according to continent for additive model (Genotype ff vs Genotype FF) of FokI polymorphism in risk of hypertension.**
(JPG)

**S7 Fig. Subgroup analysis according to continent for dominant model (Genotype ff + Ff vs Genotype FF) of FokI polymorphism in risk of hypertension.**
(JPG)

**S8 Fig. Subgroup analysis according to continent for recessive model (Genotype ff vs Genotype FF + Ff) of FokI polymorphism in risk of hypertension.**
(JPG)

**S9 Fig.** Sensitivity analysis by leave-one-out analysis for hypertension risk in Patients with FokI polymorphism (From Up to Bottom, First = Allele Model, Second = Additive Model, Third = Dominant Model, Fourth = Recessive Model).
(JPG)

**S10 Fig. Forest plot for allele model (Allele B vs Allele b) of BsmI polymorphism in risk of hypertension.**
(JPG)

**S11 Fig. Forest plot for additive model (Genotype bb vs Genotype BB) of BsmI polymorphism in risk of hypertension.**
(JPG)

**S12 Fig. Forest plot for dominant model (Genotype bb + Bb vs Genotype BB) of BsmI polymorphism in risk of hypertension.**
(JPG)

**S13 Fig. Subgroup analysis according to continent for allele model (Allele B vs Allele b) of BsmI polymorphism in risk of hypertension.**
(JPG)

**S14 Fig. Subgroup analysis according to continent for additive model (Genotype bb vs Genotype BB) of BsmI polymorphism in risk of hypertension.**
(JPG)

**S15 Fig. Subgroup analysis according to continent for dominant model (Genotype bb + Bb vs Genotype BB) of BsmI polymorphism in risk of hypertension.**
(JPG)

**S16 Fig. Subgroup analysis according to continent for recessive model (Genotype bb vs Genotype BB +Bb) of BsmI polymorphism in risk of hypertension.**
(JPG)

**S17 Fig.** Sensitivity analysis by leave-one-out analysis for hypertension risk in Patients with BsmI polymorphism (From Up to Bottom, First = Allele Model, Second = Additive Model, Third = Dominant Model, Fourth = Recessive Model).
(JPG)

**S18 Fig. Forest plot for allele model (Allele A vs Allele a) of ApaI polymorphism in risk of hypertension.**
(JPG)

**S19 Fig. Forest plot for additive model (Genotype aa vs Genotype AA) of ApaI polymorphism in risk of hypertension.**
(JPG)

**S20 Fig. Forest plot for dominant model (Genotype aa + Aa vs Genotype AA) of ApaI polymorphism in risk of hypertension.**
(JPG)

**S21 Fig. Forest plot for recessive model (Genotype aa vs Genotype AA + Aa) of ApaI polymorphism in risk of hypertension.**
(JPG)

**S22 Fig. Subgroup analysis according to continent for allele model (Allele A vs Allele a) of ApaI polymorphism in risk of hypertension.**
(JPG)

**S23 Fig. Subgroup analysis according to continent for additive model (Genotype aa vs Genotype AA) of ApaI polymorphism in risk of hypertension.**
(JPG)

**S24 Fig. Subgroup analysis according to continent for dominant model (Genotype aa + Aa vs Genotype AA) of ApaI polymorphism in risk of hypertension.**
(JPG)

**S25 Fig. Subgroup analysis according to continent for recessive model (Genotype aa vs Genotype AA + Aa) of ApaI polymorphism in risk of hypertension.**
(JPG)

**S26 Fig.** Sensitivity analysis by leave-one-out analysis for hypertension risk in Patients with ApaI polymorphism (From Up to Bottom, First = Allele Model, Second = Additive Model, Third = Dominant Model, Fourth = Recessive Model).
(JPG)

**S27 Fig. Forest plot for allele model (Allele T vs Allele t) of TaqI polymorphism in risk of hypertension.**
(JPG)

**S28 Fig. Forest plot for additive model (Genotype tt vs Genotype TT) of TaqI polymorphism in risk of hypertension.**
(JPG)

**S29 Fig. Forest plot for dominant model (Genotype tt + Tt vs Genotype TT) of TaqI polymorphism in risk of hypertension.**
(JPG)

**S30 Fig. Forest plot for recessive model (Genotype tt vs Genotype TT + Tt) of TaqI polymorphism in risk of hypertension.**
(JPG)

**S31 Fig. Subgroup analysis according to continent for dominant model (Allele T vs Allele t) of TaqI polymorphism in risk of hypertension.**
(JPG)

**S32 Fig. Subgroup analysis according to continent for additive model (Genotype tt vs Genotype TT) of TaqI polymorphism in risk of hypertension.**
(JPG)

**S33 Fig. Subgroup analysis according to continent for dominant model (Genotype tt + Tt vs Genotype TT) of TaqI polymorphism in risk of hypertension.**
(JPG)

**S34 Fig. Subgroup analysis according to continent for recessive model (Genotype tt vs Genotype TT + Tt) of TaqI polymorphism in risk of hypertension.**
(JPG)

**S35 Fig.** Sensitivity analysis by leave-one-out analysis for hypertension risk in Patients with TaqI polymorphism (From Up to Bottom, First = Allele Model, Second = Additive Model, Third = Dominant Model, Fourth = Recessive Model).
(JPG)

**S36 Fig.** Forest plot including only case-control studies for hypertension risk in patients with FokI polymorphism (From Up to Bottom, First = Allele Model, Second = Additive Model, Third = Dominant Model, Fourth = Recessive Model).
(JPG)

**S37 Fig.** Forest plot including only case-control studies for hypertension risk in patients with BsmI polymorphism (From Up to Bottom, First = Allele Model, Second = Additive Model, Third = Dominant Model, Fourth = Recessive Model).
(JPG)

**S38 Fig.** Forest plot including only case-control studies for hypertension risk in patients with ApaI polymorphism (From Up to Bottom, First = Allele Model, Second = Additive Model, Third = Dominant Model, Fourth = Recessive Model).
(JPG)

**S39 Fig.** Forest plot including only case-control studies for hypertension risk in patients with TaqI polymorphism (From Up to Bottom, First = Allele Model, Second = Additive Model, Third = Dominant Model, Fourth = Recessive Model).
(JPG)

**S40 Fig. Meta-regression for hypertension risk in patients with FokI polymorphism (Allele model) with age, Newcastle-Ottawa Scale score, and male percentage as the covariate.**
(JPG)

**S41 Fig. Meta-regression for hypertension risk in patients with FokI polymorphism (Additive model) with age, Newcastle-Ottawa Scale score, and male percentage as the covariate.**
(JPG)

**S42 Fig. Meta-regression for hypertension risk in patients with FokI polymorphism (Dominant model) with age, Newcastle-Ottawa Scale score, and male percentage as the covariate.**
(JPG)

**S43 Fig. Meta-regression for hypertension risk in patients with FokI polymorphism (Recessive model) with age, Newcastle-Ottawa Scale score, male percentage as the covariate.**
(JPG)

**S44 Fig. Meta-regression for hypertension risk in patients with BsmI polymorphism (Recessive model) with age and Newcastle-Ottawa Scale score as the covariate.**
(JPG)

**S45 Fig. Funnel plot, trim and fill analysis, with Egger and Rank Correlation test for hypertension risk in patients with FokI polymorphism (Allele model).**
(JPG)

**S46 Fig. Funnel plot, trim and fill analysis, with Egger and rank correlation test for hypertension risk in patients with FokI polymorphism (Additive model).**
(JPG)

**S47 Fig. Funnel plot, trim and fill analysis, with Egger and rank correlation test for hypertension risk in patients with FokI polymorphism (Dominant model).**
(JPG)

**S48 Fig. Funnel plot, trim and fill analysis, with Egger and rank correlation test for hypertension risk in patients with FokI polymorphism (Recessive model).**
(JPG)

**S49 Fig. Funnel plot, trim and fill analysis, with Egger and rank correlation test for hypertension risk in patients with BsmI polymorphism (Recessive model).**
(JPG)

## Author Contributions

**Conceptualization:** Iwan Dakota, Muhamad Fajri Adda'i, Rido Maulana, Ignatius Ivan, Bambang Widyantoro.

**Data curation:** Iwan Dakota, Muhamad Fajri Adda'i, Rido Maulana, Ignatius Ivan, Bambang Widyantoro.

**Formal analysis:** Iwan Dakota, Muhamad Fajri Adda'i, Rido Maulana, Ignatius Ivan, Renan Sukmawan, Bambang Widyantoro.

**Funding acquisition:** Iwan Dakota, Muhamad Fajri Adda'i, Rido Maulana, Ignatius Ivan, Renan Sukmawan, Bambang Widyantoro.

**Investigation:** Iwan Dakota, Muhamad Fajri Adda'i, Rido Maulana, Ignatius Ivan, Renan Sukmawan, Bambang Widyantoro.

**Methodology:** Iwan Dakota, Muhamad Fajri Adda'i, Rido Maulana, Ignatius Ivan, Renan Sukmawan, Bambang Widyantoro.

**Project administration:** Iwan Dakota, Muhamad Fajri Adda'i, Rido Maulana, Ignatius Ivan, Bambang Widyantoro.

**Resources:** Iwan Dakota, Muhamad Fajri Adda'i, Rido Maulana, Ignatius Ivan, Bambang Widyantoro.

**Software:** Iwan Dakota, Muhamad Fajri Adda'i, Rido Maulana, Ignatius Ivan.

**Supervision:** Iwan Dakota, Renan Sukmawan, Bambang Widyantoro.

**Validation:** Iwan Dakota, Muhamad Fajri Adda'i, Rido Maulana, Ignatius Ivan, Renan Sukmawan, Bambang Widyantoro.

**Visualization:** Iwan Dakota, Muhamad Fajri Adda'i, Rido Maulana, Ignatius Ivan.

**Writing – original draft:** Iwan Dakota, Muhamad Fajri Adda'i, Rido Maulana, Ignatius Ivan.

**Writing – review & editing:** Iwan Dakota, Muhamad Fajri Adda'i, Rido Maulana, Ignatius Ivan, Renan Sukmawan, Bambang Widyantoro.

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
