## [Decision Letter · Decision Letter 0]

15 Oct 2024

PONE-D-24-30504Association between Vitamin D Receptor Gene Polymorphism and Essential Hypertension: An Updated Systematic Review, Meta-analysis, and Meta-RegressionPLOS ONE

Dear Dr. Widyantoro,

Thank you for submitting your manuscript to PLOS ONE. After careful consideration, we feel that it has merit but does not fully meet PLOS ONE’s publication criteria as it currently stands. Therefore, we invite you to submit a revised version of the manuscript that addresses the points raised during the review process.

We look forward to receiving your revised manuscript.

Kind regards,

Laith Al-Eitan

Academic Editor

PLOS ONE

3. As required by our policy on Data Availability, please ensure your manuscript or supplementary information includes the following:

Additional Editor Comments (if provided):

Reviewers' comments:

Reviewer's Responses to Questions

**Comments to the Author**

1. Is the manuscript technically sound, and do the data support the conclusions?

Reviewer #1: Yes

Reviewer #2: Yes

2. Has the statistical analysis been performed appropriately and rigorously? 

Reviewer #1: Yes

Reviewer #2: Yes

3. Have the authors made all data underlying the findings in their manuscript fully available?

Reviewer #1: Yes

Reviewer #2: Yes

4. Is the manuscript presented in an intelligible fashion and written in standard English?

Reviewer #1: Yes

Reviewer #2: Yes

5. Review Comments to the Author

Reviewer #1: What os tge most common disease associated with this polymorphism. What about using other diseses. Why not choose other polymorphism. What fo you mean by previous review. Why include it in your study

Reviewer #2: Very well written and organized manuscript. To further improve, you may consider the following suggestions:

1.Abstract and Introduction: Consider adding an abstract that summarizes the purpose, methods, results, and conclusions. This helps readers quickly grasp the main points of your study.

2.Methodology: The methods section could benefit from additional detail. For example, specifying inclusion and exclusion criteria for the studies analyzed can enhance reproducibility.

3.Results Presentation: When presenting statistical results, consider using bullet points or tables for clearer visualization of key findings, especially for the odds ratios and confidence intervals.

4.Relate your findings to existing literature. Discuss why some polymorphisms (like BsmI) show significant associations while others do not, drawing on relevant studies.

5. It might be beneficial to discuss the limitations of your meta-analysis, such as potential publication bias or the variability of study designs among the included studies.

6. Suggest areas for further research, particularly regarding the impact of other factors on VDR polymorphisms and EH.

6. PLOS authors have the option to publish the peer review history of their article (what does this mean?). If published, this will include your full peer review and any attached files.

Reviewer #1: **Yes: **Yes

Reviewer #2: No

---

## [Author Response · Author response to Decision Letter 0]

8 Nov 2024

RESPONSE TO EDITOR (FOR SECOND EDITORIAL REVISION)

1. "It appears that your ORCiD iD has not been validated in your Editorial Manager account and we are unable to proceed until that step is complete. 

To validate your ORCiD iD in Editorial Manager, please follow the steps below: 

a. In your Editorial Manager account, please go to ‘Update my Information’ (in the upper left-hand corner of the main menu), and click on the Fetch/Validate link next to the ORCiD field. 

b. This link will take you to the ORCiD site and allow you to create a new iD or authenticate a pre-existing iD in Editorial Manager.

AUTHORS RESPONSE: We have re-link the ORCID ID

2. As required by our policy on Data Availability, please ensure your manuscript or supplementary information includes the following:

1. Thank you for including a table of studies identified in the literature search. We noted that there were only 23 studies included in this table however, your PRISMA flow chart suggests that 39 studies were assessed for eligibility. Please update this to be a numbered table of ALL studies identified in the literature search, including those that were excluded from the analyses and those that were included. Please also include headings for the table columns.

AUTHORS RESPONSE: We have revised this by attaching “Supplementary Files (Full Screening Results)”

2. Thank you for providing a table of the RAW extracted data. Please update this table to include column headings and the following information for each study:

AUTHORS RESPONSE: We have revised this by re-attaching “Supplementary Files for RAW Data” and add new file “Supplementary Files RAW Data for Replication Analyses”

RESPONSE TO EDITOR

1. PLOS ONE Style Requirements

Response: We have reviewed the PLOS ONE style guidelines and made the necessary adjustments to ensure compliance. Specifically, we have:

• Renamed the main manuscript file and all supplementary files following PLOS ONE's file naming conventions.

• Reformatted the manuscript using the PLOS ONE template, ensuring the title page, abstract, author list, affiliations, and figure legends are all in accordance with the provided guidelines.

• The revised manuscript now adheres to the style as outlined in the PLOS ONE formatting sample for the main body and title pages.

2. ORCID iD for the Corresponding Author

Response: We have ensured that the corresponding author, Bambang Widyantoro, has a valid ORCID iD. The ORCID iD for Bambang Widyantoro is already provided in the manuscript [ORCID: https://orcid.org/0000-0001-5325-4125]. Additionally, the ORCID iD has been authenticated in the Editorial Manager.

3. Data Availability

Response: We have provided a table listing all studies identified in our literature search, including the studies that were excluded. This table is included as Supplementary File RAW Data 2. For each excluded study, we have provided the reason(s) for exclusion, such as lack of genotype frequency data or ineligible study design.

Furthermore, we have created a table containing all data extracted from the included studies for the meta-analysis. 

Additionally, we have provided a risk of bias assessment using the Newcastle-Ottawa Scale (NOS) for each included study, as detailed in Supplementary Table 2-4. This table outlines the assessment of bias across all relevant domains, including study quality and certainty of evidence.

We have also clarified how missing data were handled. Specifically, missing data were addressed by contacting the authors of the original studies via email, and for studies where data could not be retrieved, we excluded them from the meta-analysis and documented this in the supplementary materials.

4. Reference List Review

Response: We have carefully reviewed the reference list and ensured that all citations are accurate and up-to-date. No retracted articles have been cited in our manuscript. Additional references were added in this revision:

4. Jiang LL, Zhang C, Zhang Y, Ma F, Guan Y. Associations between polymorphisms in VDR gene and the risk of osteoporosis: a meta-analysis. Arch Physiol Biochem. 2022;128: 1637–1644. 

5. Li L, Wu B, Liu J-Y, Yang L-B. Vitamin D receptor gene polymorphisms and type 2 diabetes: a meta-analysis. Arch Med Res. 2013;44: 235–241. 

6. Imani D, Razi B, Motallebnezhad M, Rezaei R. Association between vitamin D receptor (VDR) polymorphisms and the risk of multiple sclerosis (MS): an updated meta-analysis. BMC Neurol. 2019;19: 1–17. 

7. Xiong J, He Z, Zeng X, Zhang Y, Hu Z. Association of vitamin D receptor gene polymorphisms with systemic lupus erythematosus: a meta-analysis. Clin Exp Rheumatol. 2014;32: 174–181. 

8. Ferrer-Suay S, Alonso-Iglesias E, Tortajada-Girbés M, Carrasco-Luna J, Codoñer-Franch P. Vitamin D receptor gene ApaI and FokI polymorphisms and its association with inflammation and oxidative stress in vitamin D sufficient Caucasian Spanish children. Transl Pediatr. 2021;10: 103. 

22. Abdollahzadeh R, Shushizadeh MH, Barazandehrokh M, Choopani S, Azarnezhad A, Paknahad S, et al. Association of Vitamin D receptor gene polymorphisms and clinical/severe outcomes of COVID-19 patients. Infect Genet Evol. 2021;96: 105098. 

23. Marozik P, Rudenka A, Kobets K, Rudenka E. Vitamin D status, bone mineral density, and VDR gene polymorphism in a cohort of Belarusian postmenopausal women. Nutrients. 2021;13: 837. 

64. Touvier M, Deschasaux M, Montourcy M, Sutton A, Charnaux N, Kesse-Guyot E, et al. Determinants of vitamin D status in Caucasian adults: influence of sun exposure, dietary intake, sociodemographic, lifestyle, anthropometric, and genetic factors. J Invest Dermatol. 2015;135: 378–388. 

65. Fetahu IS, Höbaus J, Kállay E. Vitamin D and the epigenome. Front Physiol. 2014;5: 164. 

RESPONSE TO REVIEWER

Reviewer #1: 

1. What is the most common disease associated with this polymorphism? What about using other diseses. 

AUTHORS RESPONSE

We include a new paragraph in the Introduction section, addressing the broader associations of VDR polymorphisms with other diseases. 

Addition to the Introduction (after the sentence mentioning EH being linked with cardiovascular disease risks):

“Beyond essential hypertension (EH), VDR gene polymorphisms have also been linked to other common diseases such as osteoporosis, type 2 diabetes mellitus, and various autoimmune conditions, including multiple sclerosis and systemic lupus erythematosus. BsmI, FokI, ApaI, and TaqI polymorphisms have demonstrated varying degrees of association with these diseases due to their influence on vitamin D metabolism, immune modulation, and cellular differentiation. However, the specific role of VDR polymorphisms in EH has not been fully elucidated, which justifies our focused analysis on EH in this study.”

2. Why not choose other polymorphism? 

AUTHORS RESPONSE

In response to why specific polymorphisms were selected, we elaborate in the Introduction section.

Addition to the Introduction (the 4th paragraph):

“The polymorphisms chosen for this study (BsmI, FokI, ApaI, and TaqI) are the most commonly studied and clinically relevant variants in the VDR gene. Previous study has identified these polymorphisms as the genetic variants associated with blood pressure regulation through mechanisms involving the renin-angiotensin system and vascular function. Other polymorphism, such as Tru9I (rs757343), has been studied in relation to vitamin D metabolism, but evidence for their direct involvement in EH is still limited. Additionally, polymorphisms such as Cdx2 (rs11568820) are known to affect the VDR gene promoter region, but they are more commonly associated with conditions like bone density or inflammatory diseases rather than EH. Given the existing evidence and the higher prevalence of studies on these polymorphisms, we focused our analysis on the BsmI, FokI, ApaI, and TaqI variants.”

3. What do you mean by previous review? Why include it in your study?

AUTHORS RESPONSE

The “previous review” refers to earlier meta-analyses on this topic. We have provided a brief explanation of how our study builds upon and updates these reviews. 

Addition to the Discussion (before limitation discussion):

"Our study builds upon prior reviews, such as those conducted by Zhu et al. and Nunes et al., which examined the relationship between VDR polymorphisms and EH. We included additional studies published since these reviews and employed updated meta-regression techniques to assess the impact of age, sex, and study quality. This comprehensive approach allows for a more accurate understanding of the associations between VDR polymorphisms and EH risk."

Reviewer #2: 

Very well written and organized manuscript. To further improve, you may consider the following suggestions:

1. Abstract and Introduction: Consider adding an abstract that summarizes the purpose, methods, results, and conclusions. This helps readers quickly grasp the main points of your study.

AUTHORS RESPONSE

Our manuscript have contained an abstract

“The association between Vitamin D Receptor (VDR) gene polymorphisms and essential hypertension (EH) remains controversial. We searched databases (Cochrane Library, EBSCO, EMBASE, LILACS, ProQuest, PubMed, Science Direct, Springer) for studies on VDR gene polymorphisms and EH until May 30, 2024, following PRISMA guidelines. RevMan 5.4.1 provided pooled odds ratio (OR) under Hardy-Weinberg Equilibrium based on allele, additive, dominant, and recessive genetic models. Meta-regression was performed using Comprehensive Meta Analysis V3. Twenty-two studies from thirteen countries were analyzed. The recessive model suggested lower EH risk in individuals with the recessive allele (bb) of BsmI (OR: 0.81; 95%CI, 0.69 to 0.94, p=0.007; I2=35%, p=0.13). No significant associations were found for FokI, ApaI, and TaqI polymorphisms. Methodological quality significantly influenced EH risk associated with the FokI polymorphism across allele, additive, and dominant models (All p<0.0005). Male proportion influenced EH risk in the additive model for the FokI polymorphism (p=0.0235), while age impacted risk in the recessive model (p=0.0327). FokI polymorphism's influence on EH risk varies by sex, age, and study quality. BsmI polymorphism is independently associated with lower EH risk in recessive homozygotes, with no significant associations found for ApaI and TaqI polymorphisms.”

2. Methodology: The methods section could benefit from additional detail. For example, specifying inclusion and exclusion criteria for the studies analyzed can enhance reproducibility.

AUTHORS RESPONSE

We have expanded the description of the inclusion/exclusion criteria in the Methods section:

Addition to the Methods (Inclusion and Exclusion Criteria):

"The inclusion criteria for this meta-analysis were rigorously defined to ensure the selection of high-quality, relevant studies. First, we included studies that investigated adult populations, specifically individuals aged 18 years or older, who had been clinically diagnosed with EH or were undergoing treatment with antihypertensive medications. Diagnosis of EH was based on standard clinical guidelines, such as sustained systolic blood pressure (SBP) levels of 140 mmHg or higher and/or diastolic blood pressure (DBP) levels of 90 mmHg or higher. To be eligible, studies needed to investigate the association between VDR gene polymorphisms, including FokI, BsmI, ApaI, TaqI, or other SNPs, utilizing validated genotyping methods such as polymerase chain reaction (PCR), PCR-restriction fragment length polymorphism (PCR-RFLP), or PCR-TaqMan assay.

Comparisons were required to include a control group consisting of normotensive participants, defined as having SBP < 140 mmHg and DBP < 90 mmHg, with no history of hypertension or antihypertensive treatment. The primary outcome measure was the risk of EH, expressed as odds ratios (OR) with 95% confidence intervals (CIs). We included only observational studies—case-control, cohort, or cross-sectional—that reported genotype frequencies adhering to Hardy-Weinberg equilibrium (HWE), with a minimum of 10 participants per genotype to ensure sufficient statistical power.

Studies were excluded if they focused on non-hypertensive populations, including children under 18 years of age and pregnant women, due to the differences in blood pressure regulation mechanisms compared to the general adult population. We also excluded studies that failed to provide adequate data on allele and genotype frequencies or that lacked effect estimates (such as OR) for EH risk. Additionally, reviews, meta-analyses, commentaries, case reports, case series, conference abstracts, and animal studies were excluded to focus solely on observational studies with primary data.

Moreover, to maintain the integrity of the genetic analysis, studies that employed non-standard genotyping methods or failed to report quality measures for genotyping—such as PCR validation or adherence to HWE—were excluded. This ensured that only studies meeting the highest standards for genotyping quality were included in the final analysis."

3. Results Presentation: When presenting statistical results, consider using bullet points or tables for clearer visualization of key findings, especially for the odds ratios and confidence intervals.

AUTHORS RESPONSE

To improve clarity in the Results section, we have include table for the odds ratios and confidence intervals. We add a new table summarizing the key statistical findings for each polymorphism.

Table 2 (Before the section of additional analysis)

The complete summary for the risk of essential hypertension across different polymorphisms (FokI, BsmI, ApaI, and TaqI) and genetic models are available in Table 2.

Table 2. Summary of Essential Hypertension Risk in Patients with FokI, BsmI, ApaI, and TaqI Polymorphism

Polymorphism Genetic Model OR (95% CI) p-value Heterogeneity (I²)

FokI Allele model 0.96 (0.84-1.10) 0,55 78%

 Additive model 1.08 (0.82-1.44) 0,58 77%

 Dominant model 1.00 (0.84-1.18) 0,99 69%

 Recessive model 1.11 (0.86-1.44) 0,42 79%

BsmI Allele model 1.08 (0.97-1.20) 0,17 0%

 Additive model 0.91 (0.73-1.14) 0,42 0%

 Dominant model 0.98 (0.81-1.18) 0,81 0%

 Recessive model 0.81 (0.69-0.94) 0,007 35%

ApaI Allele model 1.04 (0.94-1.16) 0,45 0%

 Additive model 0.83 (0.66-1.05) 0,13 17%

 Dominant model 0.97 (0.82-1.15) 0,75 0%

 Recessive model 0.93 (0.77-1.12) 0,46 32%

TaqI Allele model 0.90 (0.67-1.21) 0,48 81%

 Additive model 1.36 (0.65-2.82) 0,41 82%

 Dominant model 1.31 (0.78-2.21) 0,31 87%

 Recessive model 0.99 (0.63-1.54) 0,95 66%

4. Relate your findings to existing literature. Discuss why some polymorphisms (like BsmI) show significant associations while others do not, drawing on relevant studies.

AUTHORS RESPONSE

In the Discussion section, we have added a paragraph that compares our findings to existing studies, particularly addressing why some polymorphisms (e.g., BsmI) show significant associations while others do not.

Addition to the Discussion (before the prior review discussion):

"The differential association between VDR polymorphisms and EH risk could be attributed to the distinct biological functions of each polymorphism. For example, the BsmI polymorphism involves a nucleotide substitution in intron 8 that influences transcript stability, potentially reducing VDR protein expression. This reduction may lead to increased RAS activity, contributing to elevated blood pressure and hypertension risk. Conversely, other polymorphisms like TaqI and ApaI, while located in similar non-coding regions, may not exert as pronounced an effect on gene expression. Studies have shown that TaqI and ApaI polymorphisms, though asso

---

## [Editor Report · Decision Letter 1]

19 Nov 2024

Association between Vitamin D Receptor Gene Polymorphism and Essential Hypertension: An Updated Systematic Review, Meta-analysis, and Meta-Regression

PONE-D-24-30504R1

Dear Dr. Bambang,

We’re pleased to inform you that your manuscript has been judged scientifically suitable for publication and will be formally accepted for publication once it meets all outstanding technical requirements.

Kind regards,

Laith Al-Eitan

Academic Editor

PLOS ONE
---

## [Editor Report · Acceptance letter]

28 Nov 2024

PONE-D-24-30504R1 

PLOS ONE

Dear Dr. Widyantoro, 

I'm pleased to inform you that your manuscript has been deemed suitable for publication in PLOS ONE. Congratulations! Your manuscript is now being handed over to our production team.

Kind regards, 

on behalf of

Professor Laith Al-Eitan 

Academic Editor

PLOS ONE